# Evaluating the Disaster Risk of the COVID-19 Pandemic Using an Ecological Niche Model

**Ping He** [1,2], **Yu Gao** [1,2], **Longfei Guo** [1,2], **Tongtong Huo** [1,2], **Yuxin Li** [3,4], **Xingren Zhang** [3,4], **Yunfeng Li** [1,2,5], **Cheng Peng** [6,*] and **Fanyun Meng** [1,2,*]

1    Beijing Key Laboratory of Traditional Chinese Medicine Protection and Utilization, Faculty of Geographical Science, Beijing Normal University, Beijing 100875, China; 201921051138@mail.bnu.edu.cn (P.H.); 201921051137@mail.bnu.edu.cn (Y.G.); 202021051122@mail.bnu.edu.cn (L.G.); 202021051125@mail.bnu.edu.cn (T.H.); 201731190020@mail.bnu.edu.cn (Y.L.)

2    Engineering Research Center of Natural Medicine, Ministry of Education, Faculty of Geographical Science, Beijing Normal University, Beijing 188875, China

3    Key Laboratory of Environmental Change and Natural Disaster, Ministry of Education, Beijing Normal University, Beijing 100875, China; lyx2019@mail.bnu.edu.cn (Y.L.); zxr@mail.bnu.edu.cn (X.Z.)

4    Academy of Disaster Reduction and Emergency Management, Ministry of Emergency Management and Ministry of Education, Beijing 100875, China

5    Key Laboratory of Research and Development of Traditional Chinese Medicine in Hebei Province, Chengde Medical College, Chengde 067000, China

6    Key Laboratory of Systematic Research of Distinctive Chinese Medicine Resources in Southwest China, Chengdu University of Traditional Chinese Medicine, Chengdu 611137, China

\*    Correspondence: pengchengchengdu@126.com (C.P.); mfy@bnu.edu.cn (F.M.); Tel.: +28-6180-0231 (C.P.); +10-5880-7656 (F.M.)

**Abstract:** Since 2019, the novel coronavirus has spread rapidly worldwide, greatly affecting social stability and human health. Pandemic prevention has become China's primary task in responding to the transmission of COVID-19. Risk mapping and the proposal and implementation of epidemic prevention measures emphasize many research efforts. In this study, we collected location information for confirmed COVID-19 cases in Beijing, Shenyang, Dalian, and Shijiazhuang from 5 October 2020 to 5 January 2021, and selected 15 environmental variables to construct a model that comprehensively considered the parameters affecting the outbreak and spread of COVID-19 epidemics. Annual average temperature, catering, medical facilities, and other variables were processed using ArcGIS 10.3 and classified into three groups, including natural environmental variables, positive socio-environmental variables, and benign socio-environmental variables. We modeled the epidemic risk distribution for each area using the MaxEnt model based on the case occurrence data and environmental variables in four regions, and evaluated the key environmental variables influencing the epidemic distribution. The results showed that medium-risk zones were mainly distributed in Changping and Shunyi in Beijing, while Huanggu District in Shenyang and the southern part of Jinzhou District and the eastern part of Ganjingzi District in Dalian also represented areas at moderate risk of epidemics. For Shijiazhuang, Xinle, Gaocheng and other places were key COVID-19 epidemic spread areas. The jackknife assessment results revealed that positive socio-environmental variables are the most important factors affecting the outbreak and spread of COVID-19. The average contribution rate of the seafood market was 21.12%, and this contribution reached as high as 61.3% in Shenyang. The comprehensive analysis showed that improved seafood market management, strengthened crowd control and information recording, industry-catered specifications, and well-trained employees have become urgently needed prevention strategies in different regions. The comprehensive analysis indicated that the niche model could be used to classify the epidemic risk and propose prevention and control strategies when combined with the assessment results of the jackknife test, thus providing a theoretical basis and information support for suppressing the spread of COVID-19 epidemics.

**Keywords:** COVID-19; risk assessment; niche model; epidemic prevention and control

## 1. Introduction

The new coronavirus disease is highly infectious and has a long incubation period with the potential for no symptoms. To date, abundant literature has indicated that the crude mortality ratio (the number of reported deaths divided by the number of reported cases) is between 2–3%, which is slightly higher than that of common influenza and lower than that of SARS [1,2]. As of 23 April 2020, 2,544,792 COVID-19 cases, including 175,694 attributable deaths, have been reported worldwide [3]. Owing to the absence of a vaccine, high viral transmission rates have occurred not only in China but also globally, significantly affecting the world's economy [4,5]. In the second half of 2020, an outbreak of the new coronavirus again seriously affected China's economic and social stability [6,7]. Since October, new coronavirus-infected persons have been successively identified in Beijing, Shenyang, Dalian, and Shijiazhuang. The top priority of researchers is to formulate epidemic-prevention measures that address local epidemic situations. In China, various regions accumulated relatively abundant epidemic-prevention and epidemic-control experience following the outbreak of the epidemic in 2019. For example, in February 2020, the Zhejiang Provincial Administration of Market Supervision issued the "Code for the Management of the Five-Color Map of Epidemic Risk Assessment" [8], which detailed the positive (total number of confirmed cases, proportion of local cases, and cluster outbreaks) and reverse (number of no new confirmed cases) indicators used to comprehensively assess the epidemic risk level in each region of the province. Sichuan Province was divided into four area types according to the actual situation: areas with no disease cases, areas with sporadic cases, areas with community outbreaks, and areas with endemic disease cases. Zones with cases for which no disease spread is indicated and zones with distributed cases correspond to low-risk and medium-risk areas, respectively, while community-outbreak areas and local epidemic areas correspond to high-risk areas [9–11]. Based on the cumulative number of confirmed cases, recent epidemic development trend, number of second-generation cases, aggregated epidemics, and other factors, Yunnan Province can be divided into three risk levels: high, medium, and low [12]. Emergency management departments can quickly judge the temporal and spatial distributions of epidemic disasters and enact corresponding measures for different risk-level zones [13–16].

However, the methods by which pandemic risk areas are delineated are inconsistent among countries and provinces, introducing some difficulties for epidemic prevention. To achieve fast and effective management in all provinces, a conventional and universal assessment method is necessary. In recent years, abundant mathematical epidemiological tools have been developed worldwide to reduce virus transmission [17–20]. Selecting the appropriate mathematical tools and formulating effective evaluation standards to conduct epidemic risk assessments and epidemic prevention for the novel coronavirus is the focus of current epidemic transmission research [21–23]. Therefore, to improve the efficiency of epidemic-prevention research, it is particularly important to develop a widely applicable assessment method to identify areas that are at risk of epidemics and curb the further transmission of the virus, simulate the risk area distributions, and map the epidemic risk. In addition, some countries are suffering from stable transmission of COVID-19 cases, while other countries are experiencing widespread transmission of the novel coronavirus. Some studies have shown that natural environmental variables play a significant role in the transmission of COVID-19, and temperature is reported as the critical factor affecting transmission [24]. Population density, precipitation conditions, human activities, and other factors may also affect the virus transmission risk [25]. Therefore, assessing the distribution of COVID-19 and understanding the impacts of environmental parameters on the spread of COVID-19 are necessary.

A species distribution model (SDM) is a tool that has been gradually applied in species distribution research and in evaluations of significant parameters [26]. The CLIMEX, DOMAIN, GARP, and MaxEnt models have been widely used to assess endangered and invasive species, parasites, and epidemics [27–32]. Research shows that the MaxEnt model has been widely used in species distribution predictions in recent years due to its simple

operation, optimal performance with a small amount of sample data, and high simulation precision, among other characteristics [33–37]. For example, Yu et al. used the MaxEnt model to predict the potential geographical distribution of the H7N9 avian influenza virus and found that Suzhou, Wuxi, and Changzhou were high-risk zones for H7N9 outbreaks [38]. Temperature and precipitation have been viewed as important environmental variables affecting the distribution of the influenza virus. Hu et al. used the niche model to simulate the habitat suitability of schistosomiasis in Yunnan, providing a powerful reference for the prevention and control of schistosomiasis transmission [39]. Li et al. analyzed the habitat suitability of dengue fever and comprehensively evaluated the environmental conditions that affect the survival of schistosomiasis virus [40]. Chaiyos et al. modeled the soil-transmitted helminth (STH) infection niche in Thailand and found that elevation and temperature played key roles in the spread of the epidemic [41]. Yu et al. used the MaxEnt model to analyze the spatial distribution of typhus in China and found that Guangdong Province is a high-incidence area and that autumn is the main season of typhus occurrence [42]. Chalghaf et al. simulated the distribution of cutaneous leishmaniasis in Tunisia by using niche models, and the results showed that temperature and precipitation had high contribution rates in the model evaluation [43]. Therefore, the MaxEnt model has become a main tool used to assess the potential geographical distributions of parasitic or epidemic species in recent years, and important environmental parameters are usually identified in these studies. For the new coronavirus disease, Coro et al. used the niche model to simulate the global distribution of the pandemic [44]. However, because of the great irrationality of the evaluation criteria, the results could not fully reveal the niche demand of the new coronavirus; this outcome has aroused the curiosity of scholars [45,46]. Ren et al. used the MaxEnt model to simulate the potential distribution of the COVID-19 epidemic in Beijing, Guangzhou, and Shenzhen at the initial stage of the epidemic outbreak [47]. Through their method, the influence of social and environmental factors in the initial stage of epidemic infection and the selected population density were found to be the most important factors affecting the distribution of the epidemic through model predictions and evaluations. However, it is difficult for their research results to have practical value regarding the proposal and implementation of epidemic prevention and control strategies. Thus, it is critical to comprehensively select appropriate parameters for regional assessments to control the source of infection and cut off the transmission route. Our study considered the influence of natural environmental variables and socio-economic variables to identify the risk distribution and derived different conclusions that provide a useful reference.

Considering Beijing, Dalian, Shenyang, and Shijiazhuang as the study area, we comprehensively selected natural and social variables affecting the outbreak and spread of COVID-19. The MaxEnt model was used to predict the potential distribution of COVID-19 and divide the risk-grade map with regard to the epidemic situation. The objectives of this study include: (1) divide the risk level of the epidemic situation by using a niche model and geographic information technology, (2) explore the important environmental factors affecting the outbreak and spread of the COVID-19 epidemic, and (3) propose prevention and control measures for the COVID-19 epidemic.

## 2. Materials and Methods

### 2.1. COVID-19 Distribution Data

Between 5 October 2020, and 5 January 2021, the number of confirmed COVID-19 cases were obtained from the official websites of the National Health Commission (http://www.nhc.gov.cn/, accessed on 5 October 2020) and the health commissions of the four cities of interest (Beijing, Dalian, Shenyang, and Shijiazhuang); these data were regarded in this study as the COVID-19 distribution record data. Because cases that were imported from abroad were discovered and quarantined sufficiently early, this aspect is not considered in this study. We used Google Earth (http://ditu.google.cn/, accessed on 20 October 2020) to obtain the latitude and longitude of each confirmed case in accordance

with the described geographical locations (individual case residences), and these data were applied as the distribution data. After screening, we finally obtained 32 COVID-19 distribution sites in Beijing, 35 sites in Shenyang, 42 sites in Dalian, and 171 sites in Shijiazhuang (Figure 1), all of which were converted into the ".csv" format.

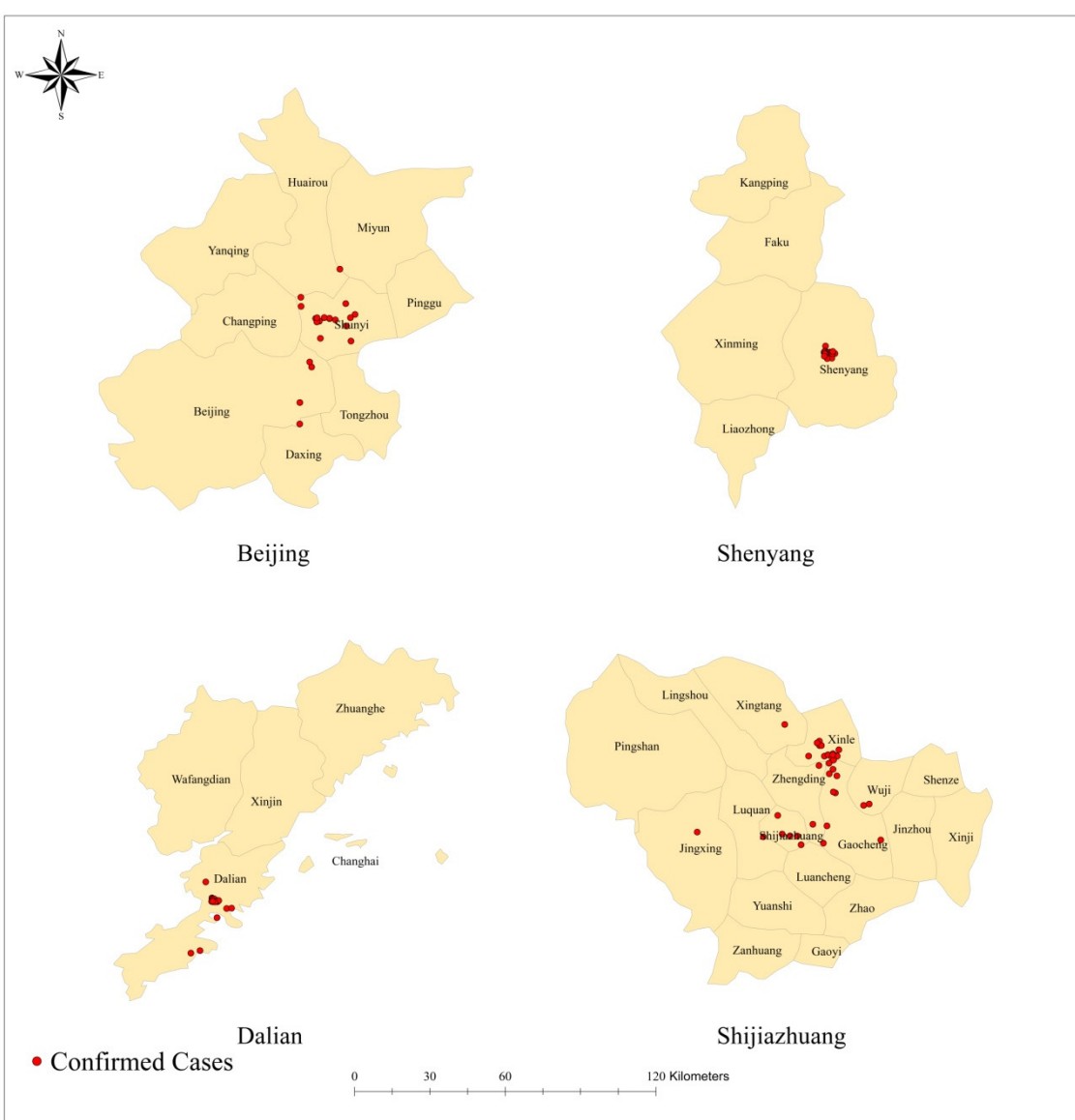

**Figure 1.** Distribution records of confirmed cases in Beijing, Shenyang, Dalian, and Shijiazhuang.

### 2.2. Environmental Parameters

To assess the effects of different kinds of environmental variables on the transmission of COVID-19, we conducted an extensive literature search to collect environmental factors that may affect the spread of the virus. We divided these environmental parameters into natural environmental variables (Group 1), positive socio-environmental variables (Group 2) and benign socio-environmental variables (Group 3). Among these groups, the natural environmental factors were obtained from the world climate database (http://worldclim.org/, accessed on 1 November 2020) and included the annual average temperature (Bio1), annual precipitation (Bio2), wettest-month precipitation (Bio3), wettest-season precipitation (Bio4), and coldest-season precipitation (Bio5). The positive socio-environmental factors included catering (Bio6), shopping (Bio7), real estate (Bio8), company businesses (Bio9), transportation (Bio10), hotels (Bio11), seafood markets (Bio12), and training institutions

(Bio13). Fever clinics (Bio14) and medical facilities (Bio15) were considered benign socio-environmental factors (Table 1). We derived these two types of socio-environmental factors from different sources. Location information for transportation, shopping, seafood markets, and fever clinics were obtained through Baidu Maps (https://map.baidu.com/, accessed on 5 November 2020), Gaode Maps (https://www.amap.com/, accessed on 5 November 2020), and Google Earth (http://ditu.google.cn/, accessed on 5 November 2020), and the rest of the data were obtained from Baidu point of interest (POI) datasets (https://www.resdc.cn/, accessed on 20 November 2020). Supermarkets that sell seafood were considered within the scope of seafood markets when collecting the location information of seafood markets. In this study, hospitals and fever clinics were regarded as two separate factors of interest because hospitals include town, county, and municipal hospitals while fever clinics are medical sites that have been established or temporarily set up for the detection of patients with fevers; these sites include clinics, community hospitals, and temporary COVID-19 detection points. To ensure the accuracy of the results, ArcGIS was used to remove redundant and unreasonable points, and the data were projected onto 1 km × 1 km gridded maps of the four study areas for the statistical analysis. Then, we used the kriging interpolation method to interpolate the socio-environmental variables into the raster layer data of the four cities, and all formats were converted into ".asc" format with an ArcMap tool (Figure 2). Finally, we obtained the spatial distributions of fifteen environmental variables in Beijing, Shenyang, Dalian, and Shijiazhuang (Appendix A). The degree of spatial clustering and influence of each variable were determined, and all factors were regarded as important variables during modeling.

**Table 1.** Basic information of the variables used in this study.

| Code | Variable | Unit | Resolution |
|------|----------|------|------------|
| Bio1 | Annual mean temperature | °C | 1 km |
| Bio2 | Annual precipitation | mm | 1 km |
| Bio3 | Wettest-month precipitation | mm | 1 km |
| Bio4 | Wettest-season precipitation | mm | 1 km |
| Bio5 | Coldest-season precipitation | mm | 1 km |
| Bio6 | Catering | / | 1 km |
| Bio7 | Shopping | / | 1 km |
| Bio8 | Real estate | / | 1 km |
| Bio9 | Company businesses | / | 1 km |
| Bio10 | Transportation | / | 1 km |
| Bio11 | Hotels | / | 1 km |
| Bio12 | Seafood markets | / | 1 km |
| Bio13 | Training institutions | / | 1 km |
| Bio14 | Fever clinics | / | 1 km |
| Bio15 | Medical facilities | / | 1 km |

*2.3. Methods*

We used MaxEnt 3.4.0 software (http://www.cs.princeton.edu/wschapire/Maxent/, accessed on 10 December 2020) to project the potential geographic distribution of COVID-19 using different distribution point information and environmental variables. After comprehensively considering the effects of different types of variables on the potential distribution COVID-19 infection risk zones, we prepared four environmental datasets for the four regions, and each environmental dataset was divided into three types: Group 1, Group 2, and Group 3. For Group 3, fever clinics (Bio14) and medical facilities (Bio15) were regarded as benign socio-environmental variables owing to the observation that while they have curbed the spread of the epidemic, they have also increased the possibility of transmission. To obtain the best accuracy, the model parameters had to be adjusted for different regions; the specific parameter settings are shown in Table 2. For example, 25% of the location point data obtained in Beijing were selected as a test set, while 75% of the occurrence records were used to train the model using the maximum possible number

of simultaneous iterations (500). In addition, we executed the MaxEnt program with 15 replicates and evaluated the averaged results. For the four regions, the remaining parameters were set to default values with a convergence threshold of $10^{-5}$, the maximum number of background points was set to 10,000, the logistic output format ".asc" was used as the output file type, and a jackknife assessment was applied to test the importance of the environmental variables.

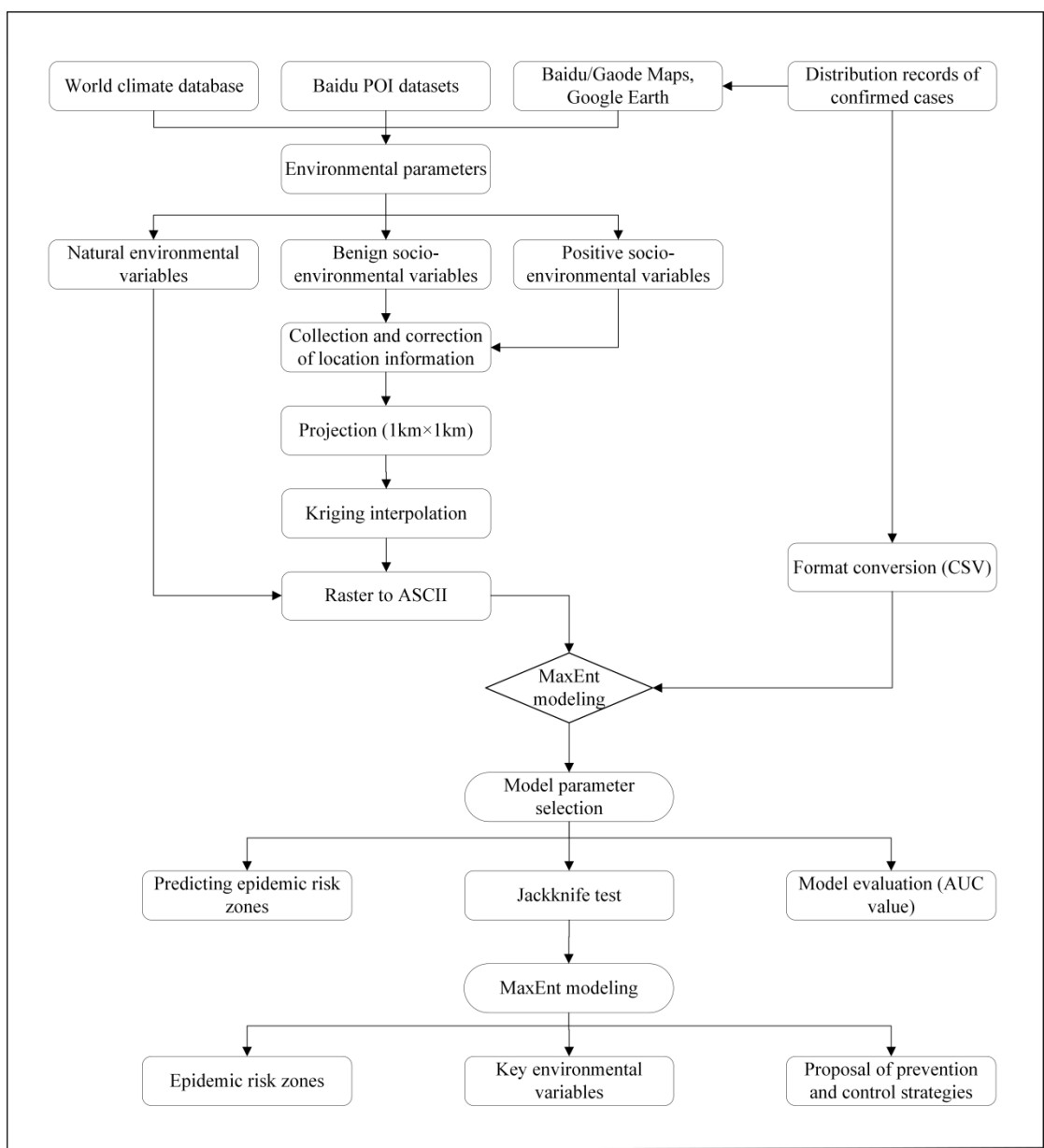

**Figure 2.** Flow diagram of the epidemic risk assessment method.

**Table 2.** Model parameter selection in different study areas.

| Area | Replicates | Maximum Iterations | Random Test Percentage (%) |
|---|---|---|---|
| Beijing | 15 | 500 | 25 |
| Shenyang | 10 | 500 | 20 |
| Dalian | 10 | 500 | 30 |
| Shijiazhuang | 10 | 1000 | 30 |

To evaluate the accuracy of the model predictions, the area under the curve (AUC) was considered [48]. In general, AUC values range from 0 to 1. When the AUC is >0.9, the modeling results are considered to reflect excellent model performance. An AUC value between 0.8 and 0.9 indicates very good model performance; a value in the range 0.7–0.8 indicates average performance; a value in the range 0.6–0.7 indicates poor performance; and a value in the range 0.5–0.6 indicates very poor performance [49,50]. Then, by combining the current distribution of risk areas across the country and through the manual classification method, we classified COVID-19 risk areas into four levels, namely, high-risk areas (0.9–1), medium-risk areas (0.4–0.9), low-risk areas (0.2–0.4) and nonrisk areas (0–0.2). Finally, we proposed some epidemic prevention and control strategies after mapping the risk distribution and identified critical environmental variables.

## 3. Results

### 3.1. Model Evaluation

We collected distribution information of confirmed COVID-19 cases and several environmental factors, such as natural environmental variables and socio-environmental factors, to construct an ecological niche model. After several runs, the results revealed that the model accuracy for Beijing was 0.933, Shenyang was 0.995, Dalian was 0.988, and Shijiazhuang was 0.895 (Figure 3). These results indicate that the model performed well and had high reliability.

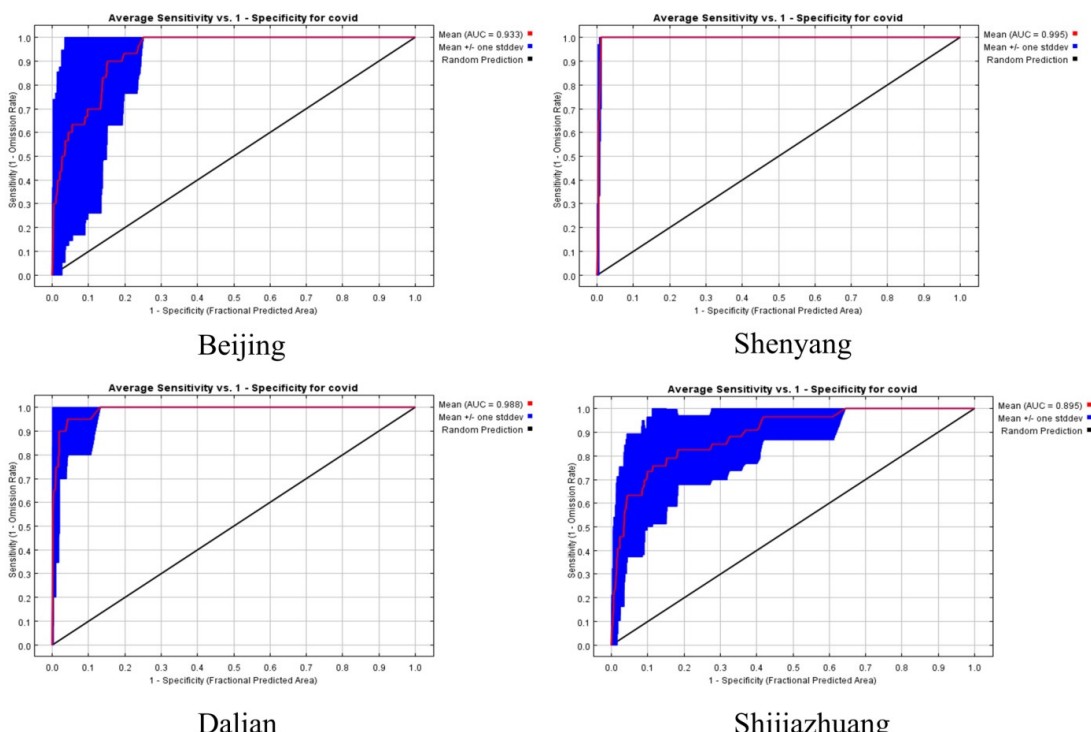

**Figure 3.** The receiver operating characteristic (ROC) curves obtained for Beijing, Shenyang, Dalian, and Shijiazhuang.

### 3.2. Epidemic Risk Level

The results reveal that from 5 October 2020 to 5 January 2021, overall, most regions were covered by nonrisk areas; the distribution of low-risk areas was small, while more medium-risk areas were identified, and high-risk areas were identified only sporadically (Figure 4). In Beijing, the medium-risk areas were mainly located at the junction of Changping, Shunyi, Huairou, and Chaoyang. Among these regions, the eastern part of Changping, near Chaoyang, is the area in which the highest epidemic risk was distributed. Low-risk zones were mainly distributed in Changping, Shunyi, and Huairou, while a few low-risk areas were distributed in Chaoyang, Tongzhou, Miyun, and Pinggu. In accordance with

the concentration of outbreaks in Shenyang, low- and medium-risk areas were mainly distributed in Huanggu District, while the distribution area of medium-risk zones was greater than that of low-risk zones. Areas in Dalian in which the epidemic risk distributions were similar to those in Shenyang were mainly concentrated in the southern part of Jinzhou and the eastern part of Ganjingzi. Among these regions, the medium-risk area was mainly distributed at the junction of the southern regions of Jinzhou and Ganjingzi, and medium-risk areas were also scattered around Shahekou District. Low-risk areas were mainly distributed near these medium-risk areas, and some other low-risk areas were located in Lvshunkou. Shijiazhuang was the area hit hardest by this COVID-19 epidemic outbreak, and the high-risk distribution zones were more scattered in this region. Medium-risk areas were mainly distributed in Xinle and Gaocheng, while most regions of Xinhua and Chang'an were also focal epidemic distribution regions. In addition to the distribution of low-risk areas around medium-risk zones, other low-risk epidemic areas were also distributed in Zhengding, Luquan, Pingshan, and Luancheng.

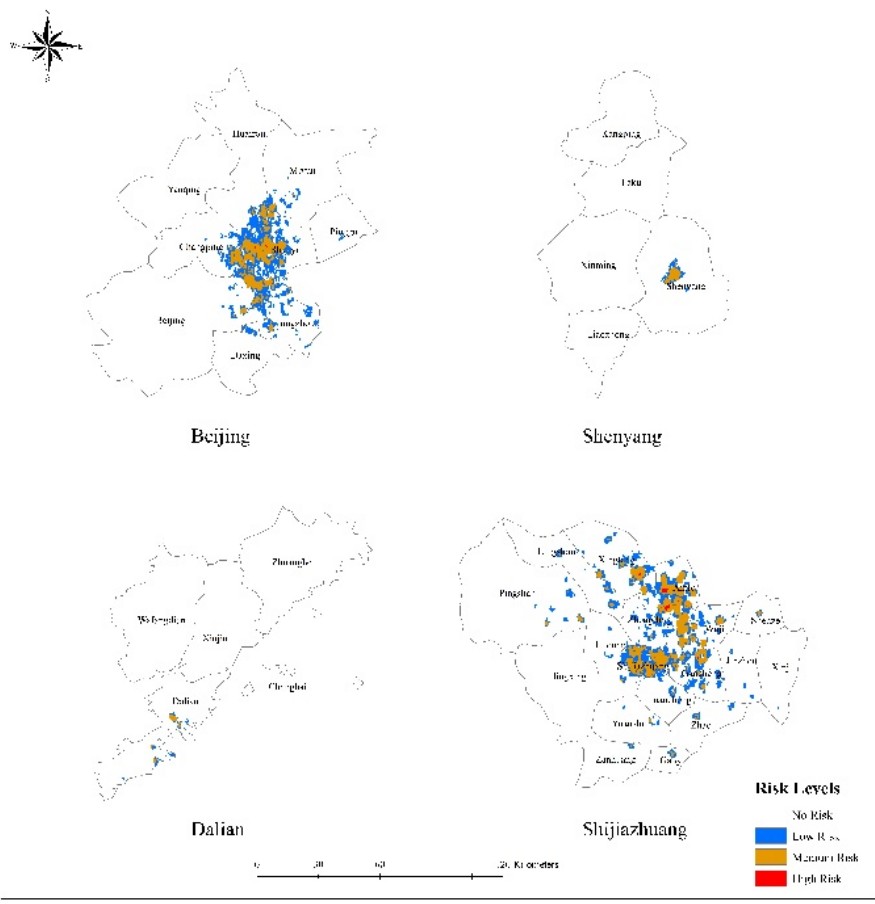

**Figure 4.** Predicted epidemic risk zones in Beijing, Shenyang, Dalian, and Shijiazhuang.

### 3.3. Key Environmental Variables

We used a jackknife test to evaluate the contribution of each considered environmental variable to the epidemic distribution, and factors with a comprehensive contribution rate > 80% and a single-factor contribution rate > 5% were identified as important environmental variables. The results showed that the shopping (Bio7), annual mean temperature (Bio1), company businesses (Bio9), wettest-month precipitation (Bio3), and seafood market (Bio12) factors were important factors driving the outbreak and continued spread of the COVID-19 epidemic in Beijing. In Shenyang, seafood markets (Bio12), training institutions (Bio13), and coldest-season precipitation (Bio5) greatly affected the outbreak and spread of the novel coronavirus. Real estate (Bio8), training institutions (Bio13), and the seafood market (Bio12) accounted for important

proportions in the risk assessment of the COVID-19 epidemic in Dalian. In Shijiazhuang, catering (Bio6), hotels (Bio11), fever clinics (Bio15), seafood markets (Bio12), coldest-season precipitation (Bio5), wettest-season precipitation (Bio4) and annual precipitation (Bio2) were considered important environmental parameters. The contribution rate of each variable is shown in Table 3.

**Table 3.** The key environmental variables and their percentage contributions in different regions.

| Beijing | Contribution (%) | Shenyang | Contribution (%) | Dalian | Contribution (%) | Shijiazhuang | Contribution (%) |
|---|---|---|---|---|---|---|---|
| Bio7 | 43.8 | Bio12 | 61.3 | Bio8 | 59.2 | Bio6 | 19.3 |
| Bio1 | 14.1 | Bio13 | 20.3 | Bio13 | 21.9 | Bio11 | 19.1 |
| Bio9 | 13.2 | Bio5 | 9.2 | Bio12 | 7.8 | Bio15 | 14.1 |
| Bio3 | 5.7 | | | | | Bio12 | 10.2 |
| Bio12 | 5.2 | | | | | Bio5 | 9.8 |
| | | | | | | Bio4 | 6.8 |
| | | | | | | Bio2 | 5.1 |

The average contribution rates of the Bio1–Bio15 factors in the models representing the four study areas were calculated to evaluate the comprehensive hazard factors influencing the regional epidemics, and factors with average total contribution rates greater than 80% were selected as important environmental factors. The results show that the seafood market (21.12%), real estate (15.62%), training institution (12.18%), shopping (11.95%), catering (5.8%), hotels (5.45%), medical (5.3%), and coldest-season precipitation (4.8%) factors substantially contributed to the model results, and the comprehensive contribution rate of these factors reached 82.2%. In addition, Group 1, Group 2, and Group 3 accounted for 16.65%, 76.33%, and 7.05%, respectively, of the total contribution rate of environmental variables (Figure 5). Positive socio-environmental variables had the greatest impact on the spread of the COVID-19 epidemics, while precipitation and temperature had lesser impacts. The influence of benign socio-environmental factors, such as the presence of medical facilities and fever clinics, was lowest because while fever clinics and medical facilities have promoted the spread of the COVID-19 epidemics, these two parameters have also played positive roles in the prevention and control of the epidemics.

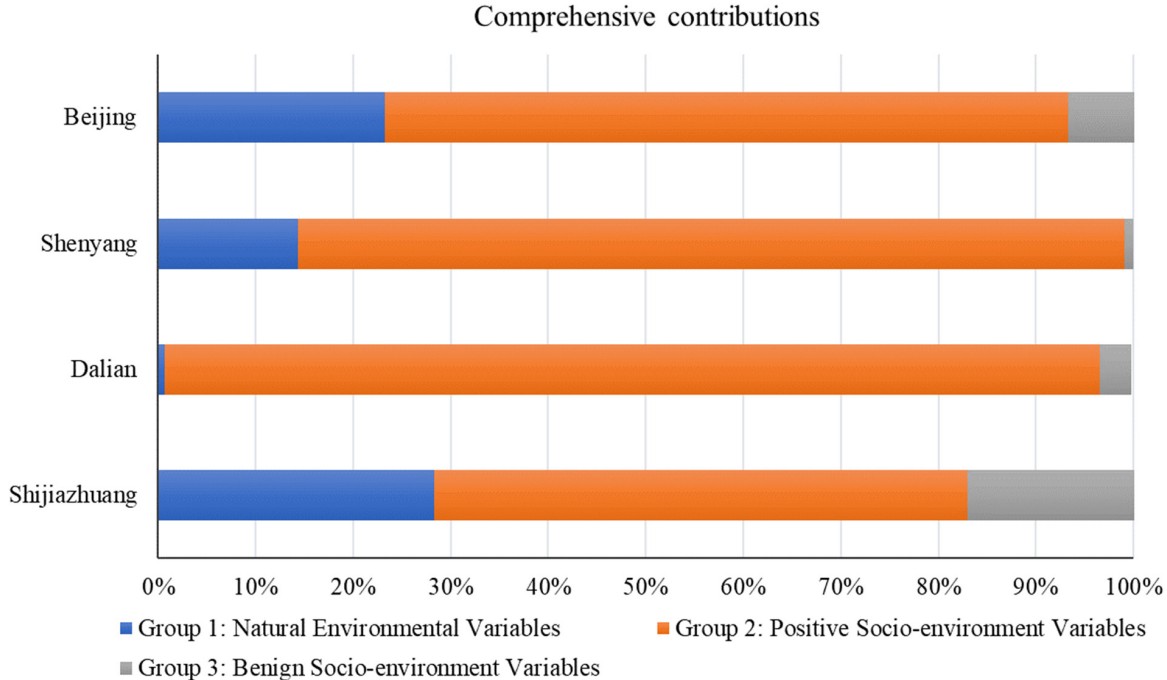

**Figure 5.** Comprehensive contribution rates of environmental variables in the four studied regions.

## 4. Discussion

To date, the COVID-19 epidemic is still ongoing in China and cannot be ignored due to its high global infection rate and mortality rate [3]. Urgent prevention and monitoring strategies must be enacted in different regions in accordance with their diverse socio-economic conditions to combat this global health threat. However, the use of different assessment methods inevitably reduces the disease management efficiency, and measures must be rapidly implemented with the arrival of this new pandemic. Herein, a universal, new, and reliable modeling method is designed to map the risk distribution and identify key parameters affecting the transmission of the virus. It is unreasonable to consider natural environmental factors alone because behavioral changes also drive the spread of infection, thus increasing the chances of the virus surviving if only natural environmental variables are considered [44,45]. We agree with Contina's statement that the underlying drivers of viral transmission are dominated by human behaviors [45], but we do not support their view that the ecological niche of SARS-CoV-2 is spurious, and our results also challenge this claim. We successfully mapped the epidemic risks in four areas and evaluated the important factors affecting viral transmission by using socio-economic and natural environmental variables. These socio-environmental parameters can represent the impacts of various factors on the spread of the virus to an excellent degree and reduce the bias introduced by the sample data or other aspects.

In our study, we combined socio-environmental factors such as supermarkets and hotels, along with natural environmental factors such as temperature and precipitation, with the occurrence distribution data of confirmed COVID-19 cases to model and evaluate the disaster risks of the COVID-19 epidemics in Beijing, Shenyang, Dalian, and Shijiazhuang from 5 October 2020 to 5 January 2021. The jackknife test method was used to evaluate the contribution rates of the factors to the spread of the COVID-19 epidemics, and all parameters were divided into natural environmental variables (Group 1), positive socio-environmental variables (Group 2) and benign socio-environmental variables (Group 3) to obtain comprehensive contribution rates. In this study, positive socio-environmental variables were found to be directly connected with the outbreak and spread of COVID-19 epidemics; this result differs from that of Ren [47], who stated that the population density is an important factor affecting the spread of the epidemic; additionally, our findings differ from those of Coro [44], who noted that temperature and precipitation are the key variables affecting the transmission of COVID-19. The results of this study allowed the variables affecting COVID-19 epidemics to be determined on a small scale and in more detail. The disaster hazard environments and hazard-causing factors differ among different regions, and emergency epidemic-prevention departments should tailor their responses according to the realistic conditions of their corresponding regions and according to the results of this model assessment. These results provide a useful reference for the prevention and control of infectious diseases through the development of prevention strategies.

According to the national epidemic risk level classification derived in January 2021, the epidemic situation was generally stable at this time. The high-risk zone was nearly nonexistent, low-risk areas and nonrisk areas accounted for large areal proportions and were all regarded as secure zones, and medium-risk zones were thus the focus of epidemic prevention and control measures. In general, the epidemic risk levels of the four studied regions basically corresponded to the risk areas obtained in this study, and because we have experience from the first round of epidemic prevention and control measures, the epidemics had short durations and small distribution areas; good control of the epidemic situations was thus achieved. For example, in Beijing, most medium-risk areas were located at the junction of Changping, Shunyi, Huairou, and Chaoyang. As the most important environmental variable assessed herein, shopping locations have characteristics of inducing high population densities, high mobilities, large crowding factors, and incomplete disinfection protection measures [51,52]. On the one hand, as the capital of China, Beijing has a very large flow of people [53,54], and supermarkets and shopping malls are urban consumption centers; thus, it is difficult to implement epidemic prevention measures in

this region. Zou et al. [51] and Ren et al. [47] also believe that population density is an important variable affecting the spread of the COVID-19 epidemic. Many studies have shown that population size and the speed and area of population movements greatly affect the spread of COVID-19 epidemics and increase the infection rates of vulnerable groups such as elderly individuals and students [53,55–58]. On the other hand, supermarkets are concentrated areas of refrigerated seafood, and these markets are stocked with large amounts of frozen meat, fish, and fruit materials, all of which increase the possibility of COVID-19 transmission. In addition, the annual mean temperature and the wettest-month precipitation month have also greatly affected the spread of the epidemic in Beijing. A large number of studies have found that the survival rate of COVID-19 increases sharply in low-temperature environments and that its survival temperature ranges from 5–11 °C [59–63]. Therefore, low temperature and precipitation conditions have played a key role in the spread of the epidemic in this region.

Shenyang and Dalian have the highest-accuracy model simulations because the distributions of confirmed cases in these two areas are relatively concentrated, leading to a higher degree of model fitting. Huanggu in Shenyang and Jinzhou (Jinpu New District) in Dalian are important economic centers and are also the main areas in which medium-risk zones are distributed, although the areas of these zones are small. For Shenyang and Dalian, the positive socio-environmental factors (Group 2) contributed more than 90% of the model results, and the most important parameters in the two regions were seafood markets and real estate. Studies have shown that, as the provincial capital and due to its vicinity to the sea, Shenyang is a key area for import and export trade activities, which generate many transactions [64–66]. Frozen food from abroad is prone to carrying COVID-19, and the presence of a large number of marine products further increases the risk of outbreaks [67,68]. Training institutions reflect the development levels of these two regions, and elderly individuals and students are vulnerable to COVID-19 infection due to their weak resistance. As a coastal city, the management of personnel flow is inherently a weak link, and if the epidemic prevention is lenient, COVID-19 can spread considerably. In Shijiazhuang, Gaocheng is the disaster zone of the outbreak, and catering, hotels, and other industries are distributed in the high-risk areas. This result reflects the higher population densities and population mobilities in these areas. Additionally, the catering industry has always been associated with food safety issues, as large amounts of seafood and frozen meat are imported for use in this industry. At the same time, employees in this region represent a high-risk group [69].

Generally, positive socio-environmental parameters have played a key role in the spread of COVID-19. Natural environmental variables such as temperature and precipitation have accelerated the spread of the epidemic, while benign socio-environmental factors such as medical facilities and fever clinics have played a role in suppressing COVID-19. As northern cities, Shijiazhuang and Shenyang are provincial capitals, Beijing is the national capital, and Dalian is close to the Yellow Sea and the Bohai Sea. In recent years, several areas in these cities have developed rapidly. Multiple factors have placed tremendous pressure on the prevention and control of COVID-19 epidemics in these cities: the populations are becoming more active; the economy and trade are becoming more dynamic; and the temperature in the northern region of China decreases after October, after which a cold wave gradually approaches, indoor ventilation is reduced and the immune system-based resistance of citizens declines [70–72]. In the face of these controllable and uncontrollable factors, epidemic-prevention departments can adopt different prevention and control strategies in consideration of epidemic risk zones and important environmental factors in various regions under the context of national guidelines and local policies. For Beijing, (1) disease surveillance, crowd control in supermarkets, and food and customs surveillance efforts need to be strengthened. (2) The sanitation emergency plan needs to be improved and followed up on, and the staffs of various units need to be closely and continuously monitored, especially those who are in close contact with seafood or refrigerated food, to prevent the occurrence of infections. (3) Since natural environmental variables have played

a certain role in the model evaluations conducted herein, we should guide residents to consider proper indoor ventilation in autumn and winter and raise residents' awareness of the "early detection, early reporting, early isolation, and early treatment" of illness. For Shenyang and Dalian, the appropriate parties should (1) closely cooperate with management departments to strengthen the monitoring of the seafood market, strengthen border security (such as by requiring quarantine upon entry), and accurately record purchase information while controlling the flow of people. The management and training of vendors in performing adequate sanitation and disinfection is another focus of our work. (2) While strictly controlling population flows, especially of companies and enterprises, related departments should strengthen epidemic prevention work after in-person work resumes; these prevention measures could include body temperature testing, ventilation, and disinfection. In the Shijiazhuang area, relevant parties should (1) strengthen their management and monitoring of the catering industry, including at restaurants and hotels, to increase the detection of refrigerated food entering the market. Staff training and protection work must be implemented to ensure their own safety and the safety of handled food. (2) For hotel and guesthouse occupants, it is necessary to conduct temperature monitoring, control the flow of people, and conduct personal information registration and disinfection measures. (3) The seafood market must be actively managed, and personal protection must be applied in winter.

## 5. Conclusions

The niche model is a mathematical tool that is widely used in many fields. Studies have shown that the niche model can be successfully used in the fields of biological invasion, endangered species protection, traditional Chinese medicine, archaeology, urban planning, and epidemiology. Our study used the MaxEnt model, which is considered the best-performing niche model, to simulate the potential geographic distribution of COVID-19 epidemics in four regions, evaluate the important variables affecting the spread of these epidemics, and propose epidemic prevention and control strategies. The results represent precise predictions and provide a new division method for epidemic risk zones, thus providing theoretical support for future epidemic prevention and control measures. Our study provides an excellent foundation for epidemic prevention work on a fine scale and for conducting effective management activities; this study not only represents the development of an extensive mathematical modeling method for China but also has positive significance for the optimization of epidemic prevention management in different countries worldwide.

This study also has some shortcomings. For example, the diversity of the niche model may affect the accuracy of the model evaluation results. Second, since humans are a very active species, human behaviors affect the spread of COVID-19 epidemics, and both social and bioclimatic conditions vastly influence the spread of COVID-19 epidemics and complicate relevant assessments. Due to the scarcity of deaths caused by these epidemics, we used confirmed COVID-19 cases as the occurrence records in this study. If large-scale mortality occurred, we would need to further assign a certain weight to these data when building the model. Finally, the number of clustered cases, the cumulative number of confirmed cases, and the number of cured cases also affect the collection of recorded confirmed-case data points and thus affect the model results. Our work will continue to focus on changes in the temporal and spatial patterns of COVID-19 to improve the accuracy of model predictions. We hope this study can provide useful references for future epidemic prevention research.

**Author Contributions:** P.H.: conceptualization, validation, writing—original draft. This author should be considered first author. Y.G. and L.G.: investigation; T.H.: data curation. The authors contributed equally to this paper and its review, and should be considered second authors. The following authors made the same contributions in this paper and should be considered third authors: Y.L. (Yuxin Li), X.Z. and Y.L. (Yunfeng Li), C.P. and F.M. reviewed and provided critical comments. All authors have read and agreed to the published version of the manuscript.

**Funding:** This work is funded by the National Natural Science Foundation of China (Grant No. 81072999) and Project of Hebei Administration of Traditional Chinese Medicine (Grant No. 2017079).

**Acknowledgments:** We wish to express our gratitude to the reviewers of this paper. They provided useful feedback on an earlier version of the manuscript.

**Conflicts of Interest:** All authors declare no competing interests.

## Appendix A

The 15 natural environment variables used in the four study zones. Among these variables, Bio1–Bio5 are natural environmental factors derived from the World Climate Database (http://worldclim.org/, accessed on 1 November 2020). Bio6–Bio15 are socio-environmental variables obtained by manual collection and kriging interpolation.

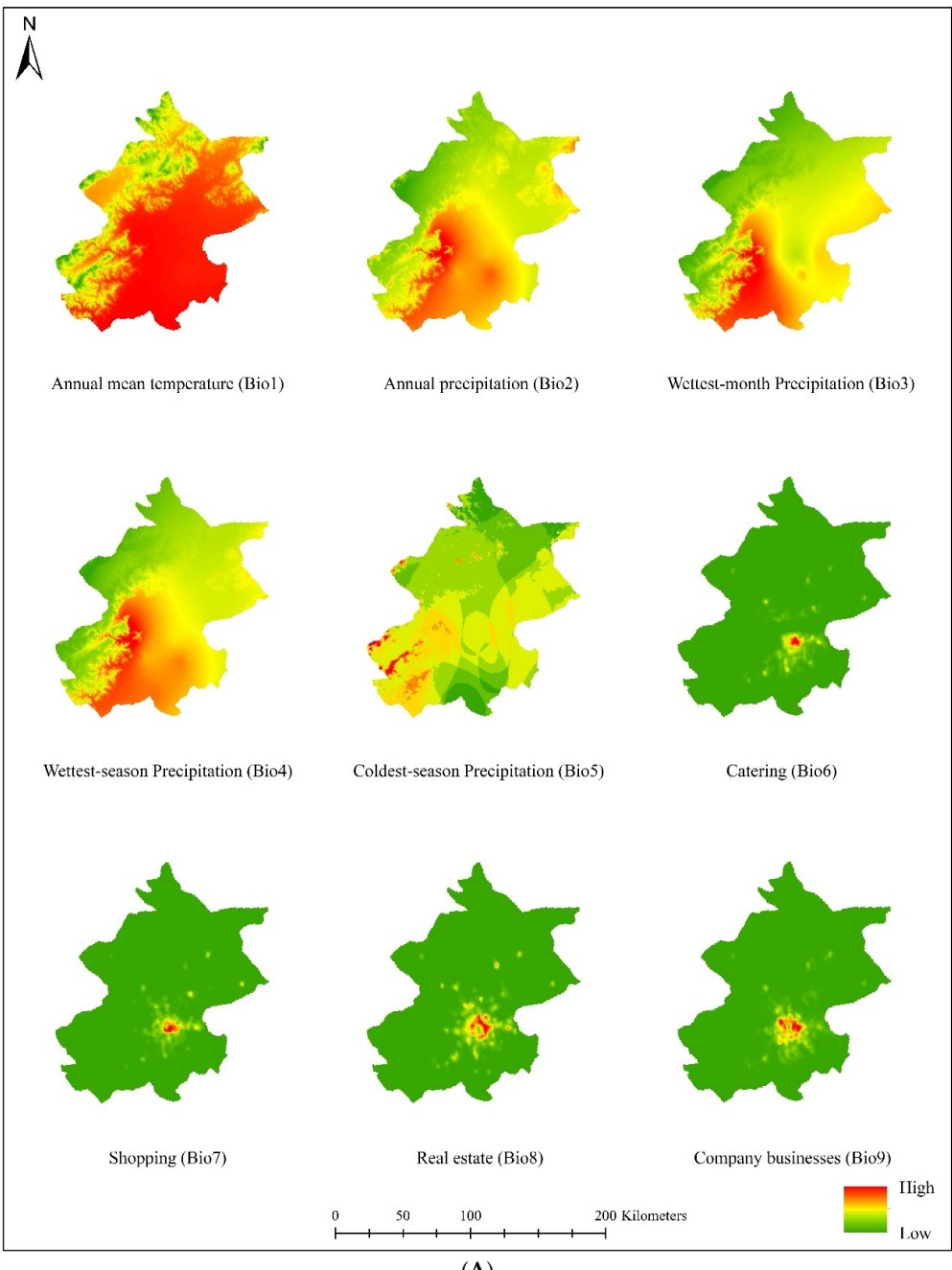

**Figure A1.** *Cont.*

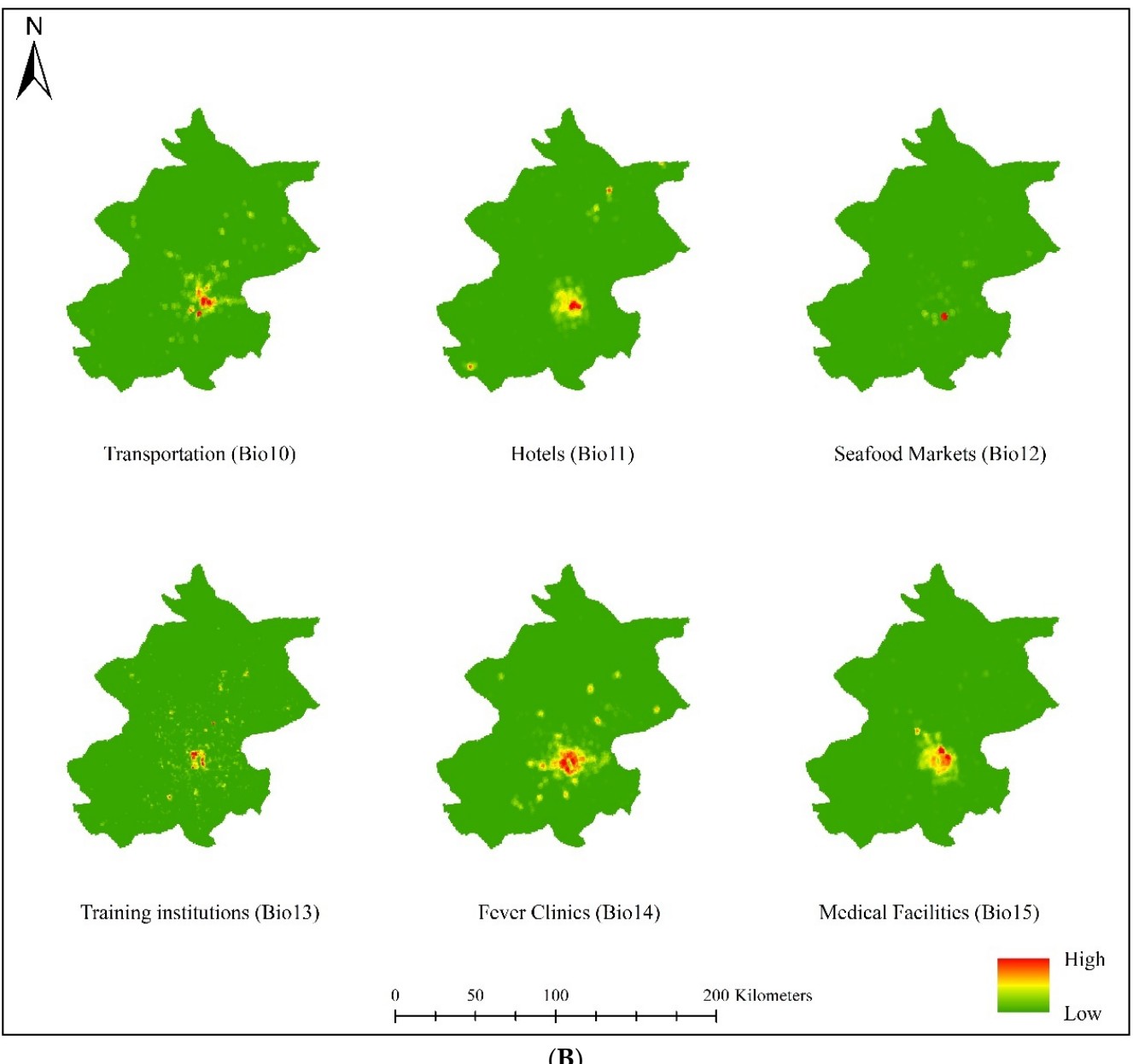

(**B**)

**Figure A1.** The 15 environment variables considered in Beijing. (**A**) the variables Bio1-Bio9; (**B**) the variables Bio10-Bio15.

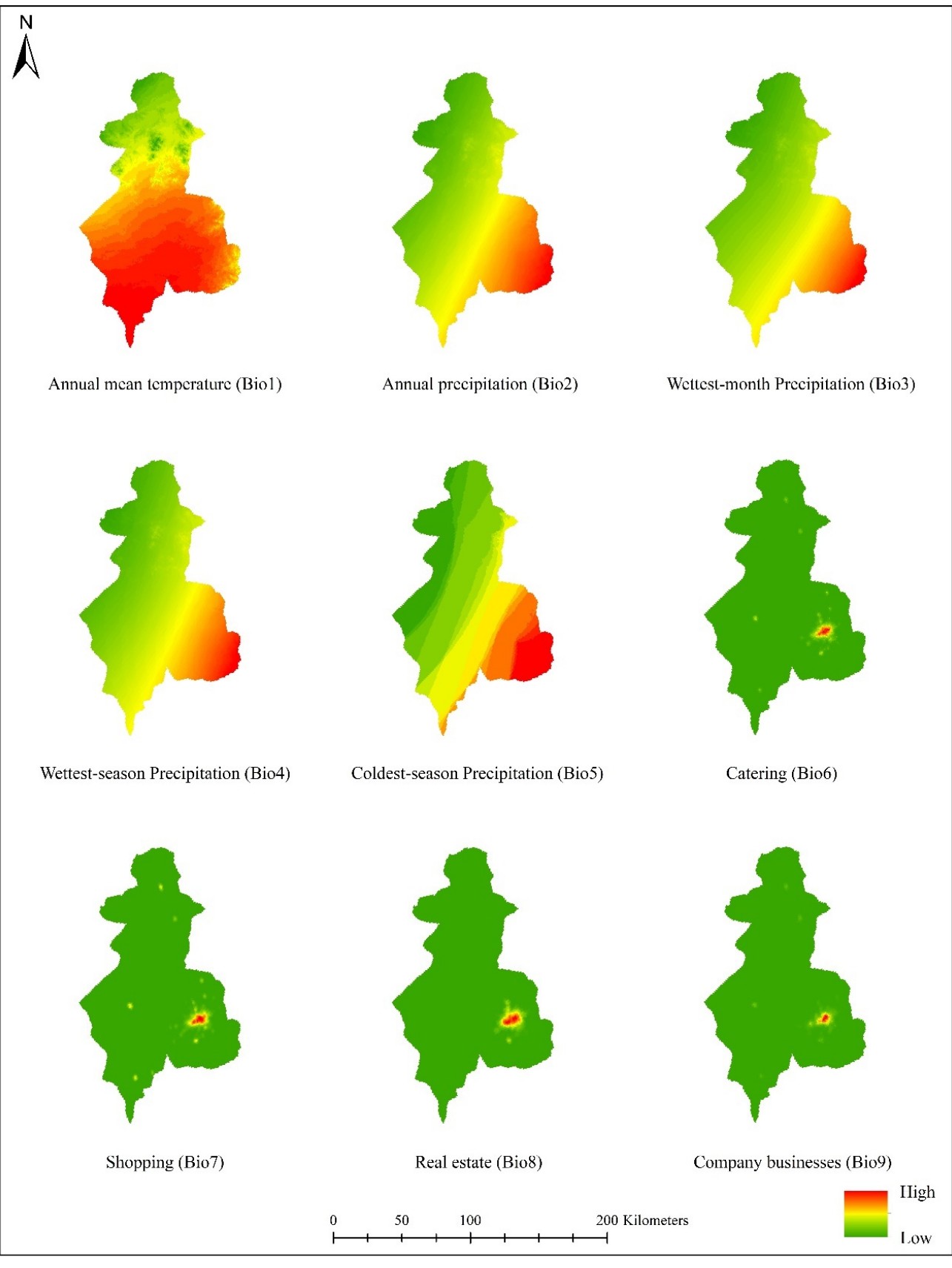

**(A)**

**Figure A2.** *Cont.*

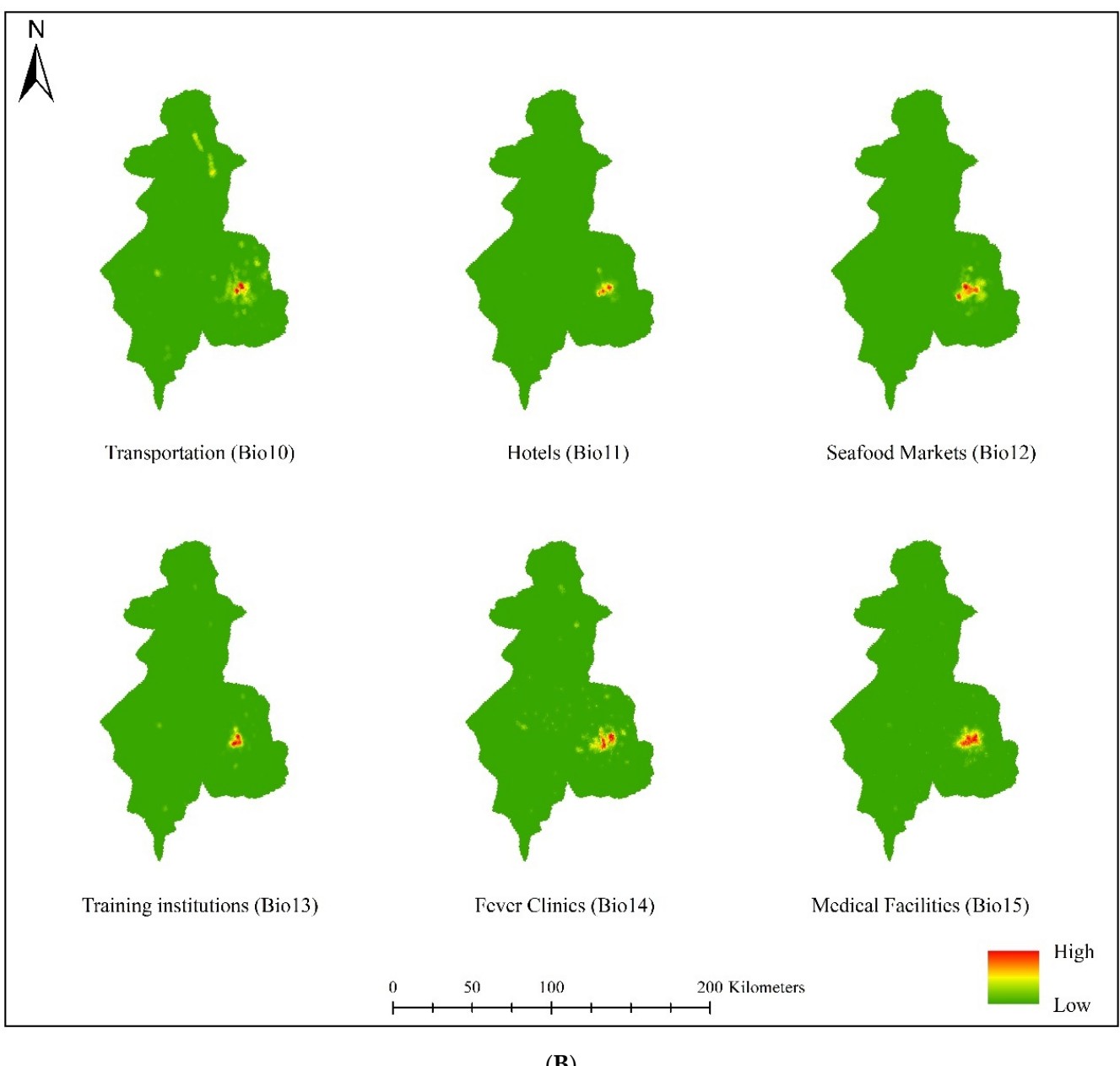

**(B)**

**Figure A2.** The 15 environment variables considered in Shengyang. (**A**) the variables Bio1-Bio9; (**B**)the variables Bio10-Bio15.

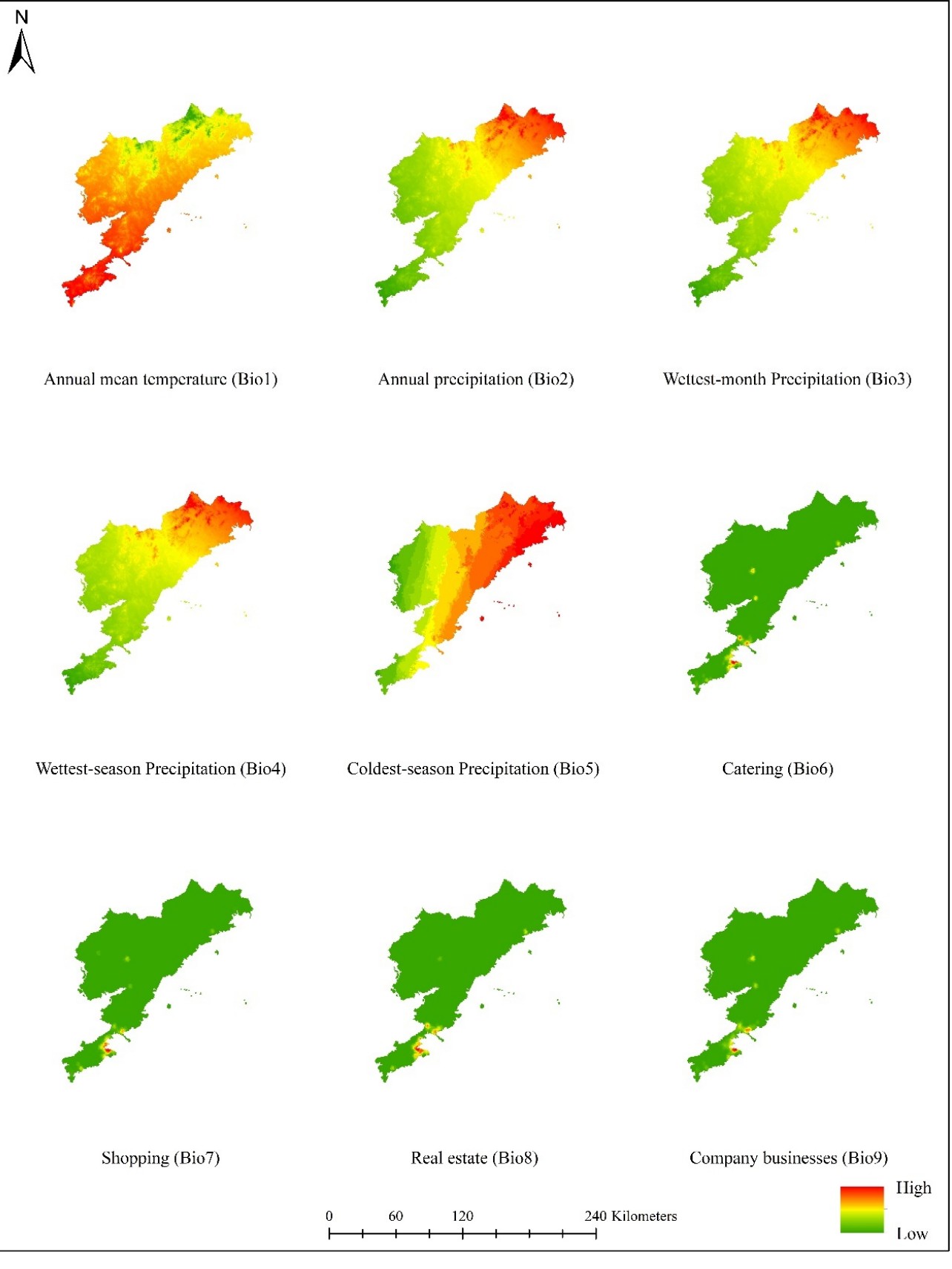

(A)

**Figure A3.** *Cont.*

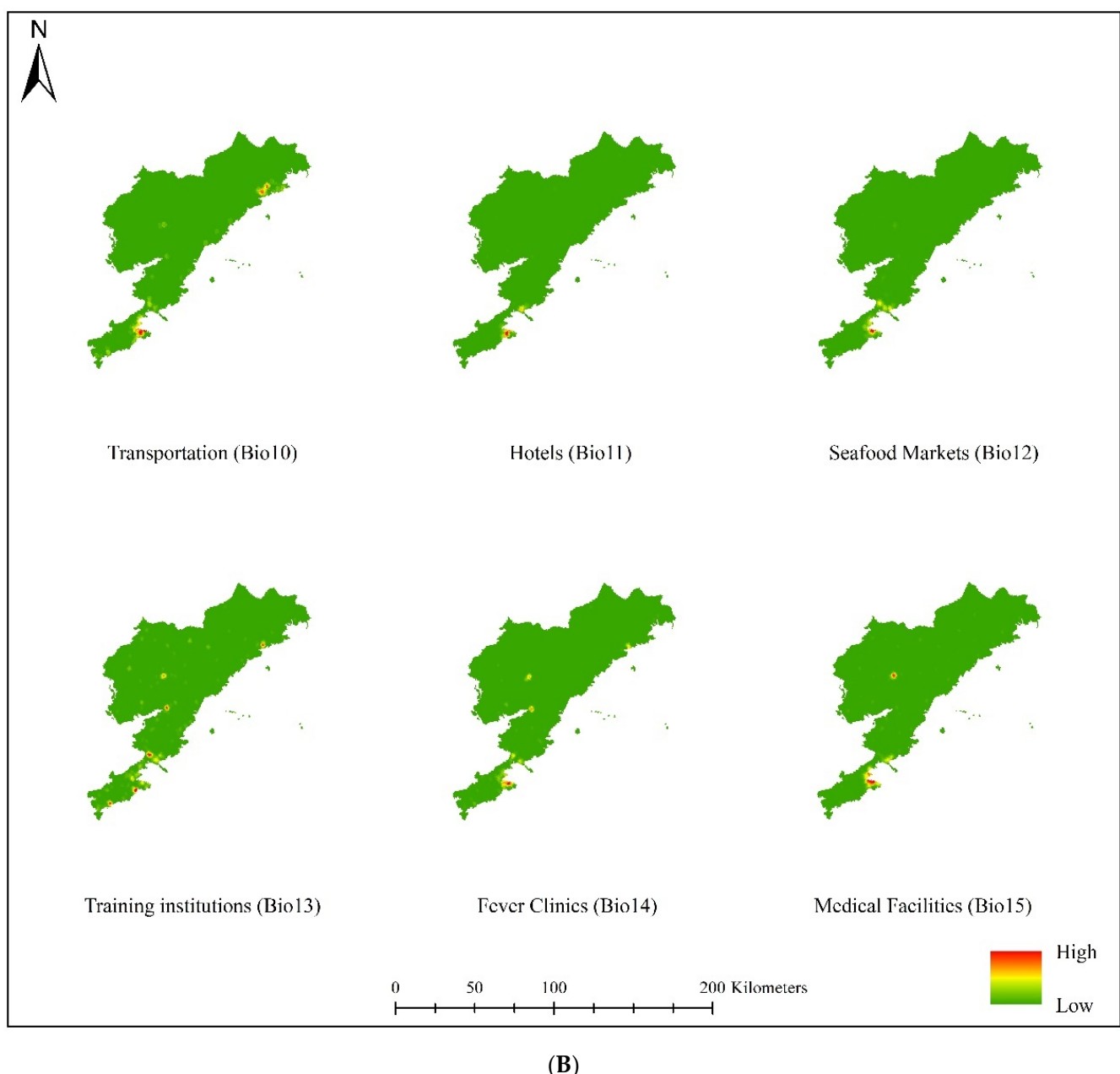

(**B**)

**Figure A3.** The 15 environment variables considered in Dalian. (**A**) the variables Bio1-Bio9; (**B**)the variables Bio10-Bio15.

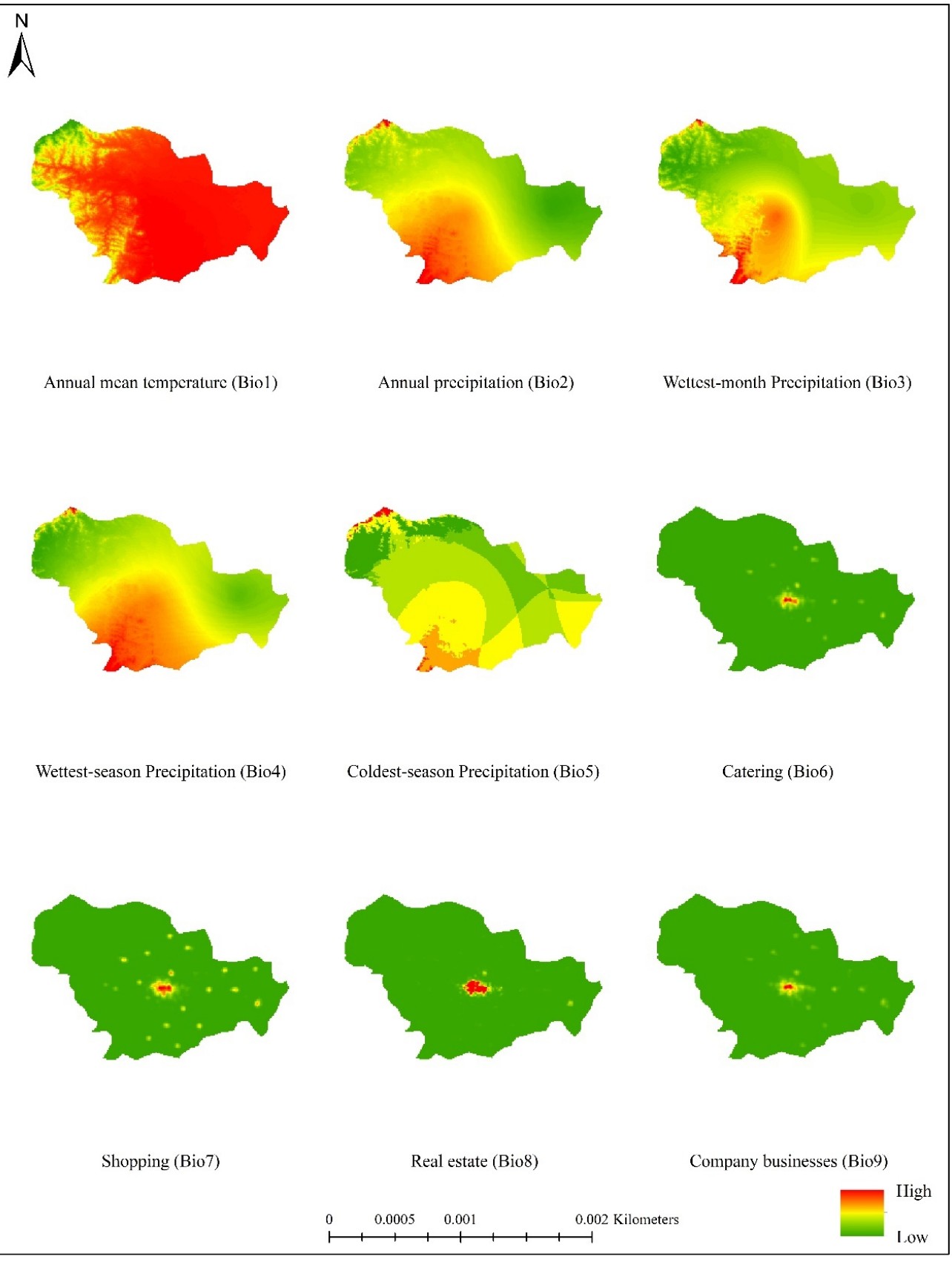

**(A)**

**Figure A4.** *Cont.*

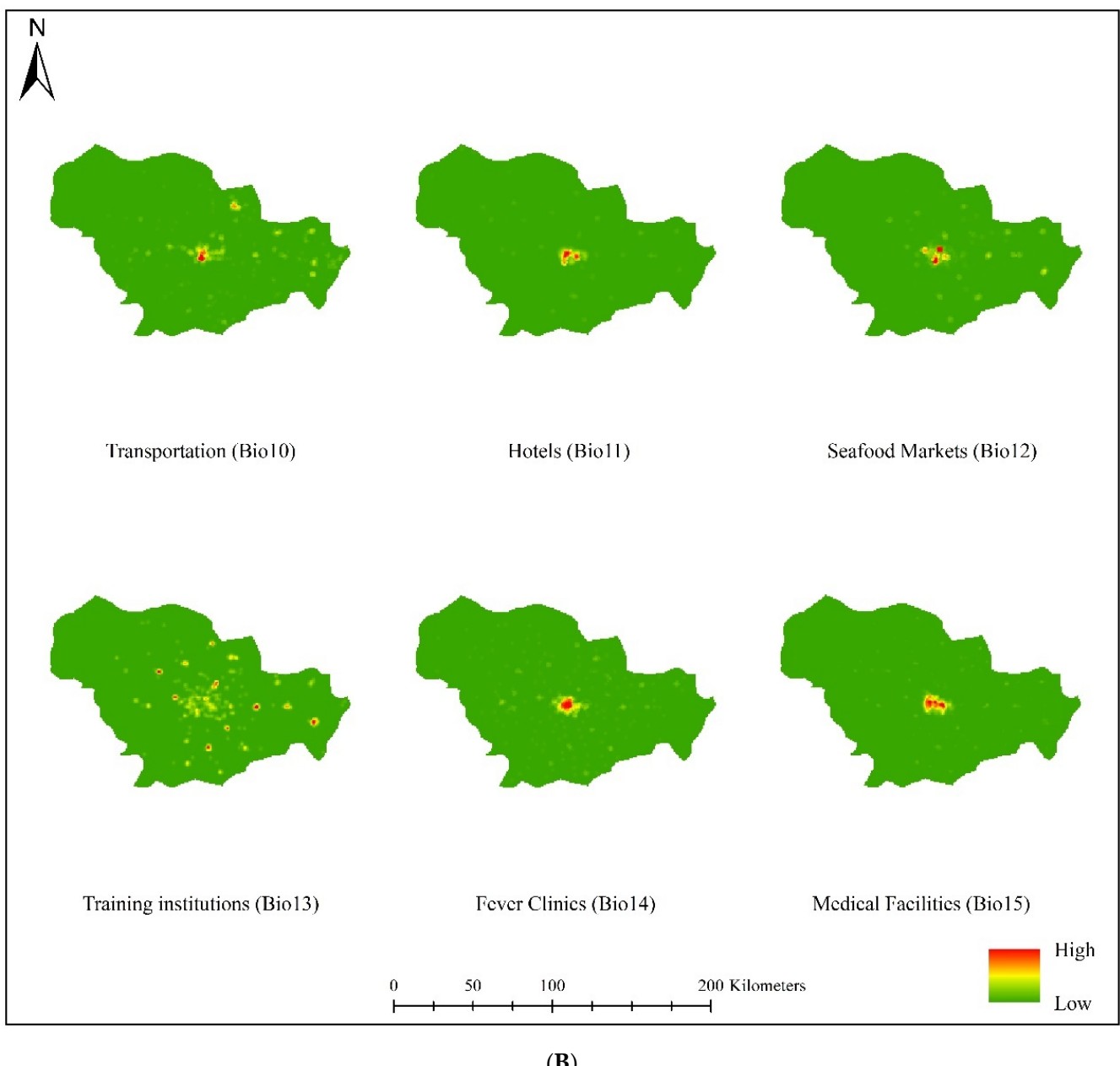

**(B)**

**Figure A4.** The 15 environment variables considered in Shiajizhuang. (**A**) the variables Bio1-Bio9; (**B**)the variables Bio10-Bio15.

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
