# Peer review of "Evaluating the Disaster Risk of the COVID-19 Pandemic Using an Ecological Niche Model"

_sustainability, doi:10.3390/su132111667_

Round 1

Reviewer 1 Report

The main research question is not well formulated nor addressed.

Outbreak of COVID-19, jackknife analysis– I suggest to rethink these expressions.

Baidu POI – what  acronym is it?

In Abstract, description of method is not finished. Collecting information and specification of variables are not enough.

What method of prediction was applied? Please, describe.

The other group of Figures 1-4 should be in the paper text and they should be interpreted.

It is very unclear why authors used the same numbers for different figures.

The paper is not relevant, authors do not consider the sustainability.

Authors should explain applicability of proposed approach for data from other countries,

Authors are requested to explain the Covid -19 situation in 2021, if the considered model of predication is applicable. If not, authors should explain why

Authors do not explain what the research adds to the subject area compared with other published material,

The paper is not well written. The paper has too many authors, and lack of coordination of their work

It's very poor paper,  so as such I can' t recommend for publishing.

Reviewer 2 Report

The content of the paper is good and the idea behind is original.

However, I have some suggestions.

  • In conclusion, the authors say more precise, but precise compared to what? The authors have to compare their numerical results to the other published ones.
  • A comparison and discussion to other approaches especially the ones in references [36] and [37], is highly recommended.
  • Section 2.3. is poor, I suggest to move Figure 2. to this section with comments and more technical clarification of the steps in the diagram, in order to not let the reader thinks this was only about using an already existing software.

Reviewer 3 Report

Some minor input could complete the already good work

Round 2

Reviewer 1 Report

Authors made a lot of improvements in comparison with the last version. I would suggest to move the figures (which are at the end) to the paper or just reject. It is required to descrbe them. 

Reviewer 2 Report

The method and conclusion sections have become now clearer and interesting.

After this revision, the content of the manuscript deserves to be read and accepted.

Thus, I recommend the publication of this work in Sustainability.